# ENHANCING HIGH-RESOLUTION 3D GENERATION THROUGH PIXEL-WISE GRADIENT CLIPPING

**Zijie Pan[1], Jiachen Lu[1], Xiatian Zhu[2], Li Zhang[1]***
[1]School of Data Science, Fudan University     [2]University of Surrey

https://fudan-zvg.github.io/PGC-3D/

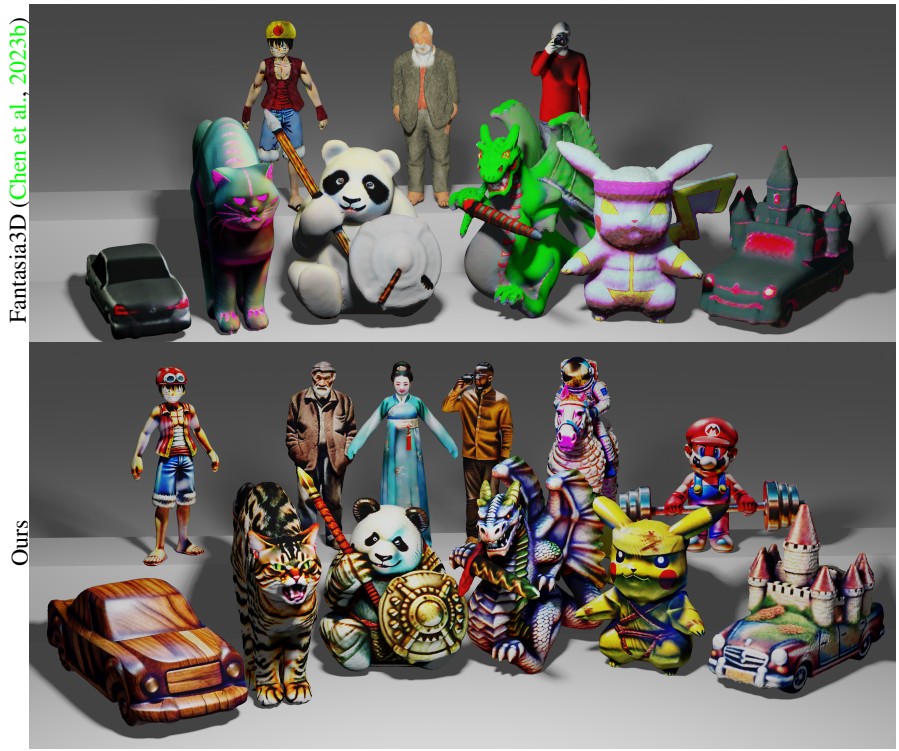

Figure 1: **Blender rendering for textured meshes. Top:** Fantasia3D (Chen et al., 2023b). **Bottom:** Ours. For each mesh in the top, we can find a corresponding one in the bottom whose texture is generated conditioned on the same prompt. Our method generates more detailed and realistic texture and exhibits better consistency with input prompts.

## ABSTRACT

High-resolution 3D object generation remains a challenging task primarily due to the limited availability of comprehensive annotated training data. Recent advancements have aimed to overcome this constraint by harnessing image generative models, pretrained on extensive curated web datasets, using knowledge transfer techniques like Score Distillation Sampling (SDS). Efficiently addressing the requirements of high-resolution rendering often necessitates the adoption of latent representation-based models, such as the Latent Diffusion Model (LDM). In this framework, a significant challenge arises: To compute gradients for individual image pixels, it is necessary to backpropagate gradients from the designated latent space through the frozen components of the image model, such as the VAE encoder used within LDM. However, this gradient propagation pathway has never been optimized, remaining uncontrolled during training. We find that the unregulated gradients adversely affect the 3D model's capacity in acquiring texture-related information from the image generative model, leading to poor quality appearance synthesis. To address this overarching challenge, we propose an innova-

---

*Li Zhang (lizhangfd@fudan.edu.cn) is the corresponding author.

tive operation termed *Pixel-wise Gradient Clipping* (PGC) designed for seamless integration into existing 3D generative models, thereby enhancing their synthesis quality. Specifically, we control the magnitude of stochastic gradients by clipping the *pixel-wise* gradients efficiently, while preserving crucial texture-related gradient directions. Despite this simplicity and minimal extra cost, extensive experiments demonstrate the efficacy of our PGC in enhancing the performance of existing 3D generative models for high-resolution object rendering.

# 1 INTRODUCTION

Motivated by the success of 2D image generation (Ho et al., 2020; Rombach et al., 2022), substantial advancements have occurred in conditioned 3D generation. One notable example involves using a pre-trained text-conditioned diffusion model (Saharia et al., 2022), employing a knowledge distillation method named "score distillation sampling" (SDS), to train 3D models. The goal is to align the sampling procedure used for generating rendered images from the Neural Radiance Field (NeRF) (Mildenhall et al., 2021) with the denoising process applied to 2D image generation from textual prompts.

However, surpassing the generation of low-resolution images (e.g., 64×64 pixels) presents greater challenges, demanding more computational resources and attention to fine-grained details. To address these challenges, the utilization of latent generative models, such as the Latent Diffusion Model (LDM) (Rombach et al., 2022), becomes necessary as exemplified in (Lin et al., 2022; Chen et al., 2023b; Tsalicoglou et al., 2023; Wang et al., 2023; Zhu & Zhuang, 2023; Hertz et al., 2023).

Gradient propagation in these methods comprises two phases. In the initial phase, gradients propagate from the latent variable to the rendered image through a pre-trained and frozen model (e.g., Variational Autoencoder (VAE) or LDM). In the subsequent phase, gradients flow from the image to the parameters of the 3D model, where gradient regulation techniques, such as activation functions and L2 normalization, are applied to ensure smoother gradient descent. Notably, prior research has overlooked the importance of gradient manipulation in the first phase, which is fundamentally pivotal in preserving texture-rich information in 3D generation.

We contend that neglecting pixel-wise gradient regulation in the first phase can pose issues for 3D model training and ultimate performance since pixel-wise gradients convey crucial information about texture, particularly for the inherently unstable VAE with the latest SDXL (Podell et al., 2023), used for image generation at the resolution of 1024×1024 pixels, as illustrated in the second column of Figure 2. The pronounced presence of unexpected noise pixel-wise gradients obscures the regular pixel-wise gradient, leading to a blurred regular gradient. Consequently, this blurring effect causes the generated 3D model to lose intricate texture details or, in severe cases of SDXL (Podell et al., 2023), the entire texture altogether.

Motivated by these observations, in this study, we introduce a straightforward yet effective variant of gradient clipping, referred to as Pixel-wise Gradient Clipping (PGC). This technique is specifically tailored for existing 3D generative models. Concretely, PGC truncates unexpected pixel-wise gradients against predefined thresholds along the pixel vector's direction for each individual pixel. Theoretical analysis demonstrates that when the clipping threshold is set around the bounded variance of the pixel-wise gradient, the norm of the truncated gradient is bounded by the expectation of the 2D pixel residual. This preservation of the norm helps maintain the hue of the 2D image texture and enhances the overall fidelity of the texture. Importantly, PGC seamlessly integrates with existing SDS loss functions and LDM-based 3D generation frameworks. This integration results in a significant enhancement in texture quality, especially when leveraging advanced image generative models like SDXL (Podell et al., 2023).

Our **contributions** are as follows: **(i)** We identify a critical and generic issue in optimizing high-resolution 3D models, namely, the unregulated pixel-wise gradients of the latent variable against the rendered image. **(ii)** To address this issue, we introduce an efficient and effective approach called Pixel-wise Gradient Clipping (PGC). This technique adapts traditional gradient clipping to regulate pixel-wise gradient magnitudes while preserving essential texture information. **(iii)** Extensive experiments demonstrate that PGC can serve as a generic integrative plug-in, consistently benefiting existing SDS and LDM-based 3D generative models, leading to significant improvements in high-resolution 3D texture synthesis.

## 2 RELATED WORK

**2D diffusion models**  Image diffusion models have made significant advancements (Ho et al., 2020; Balaji et al., 2022; Saharia et al., 2022; Rombach et al., 2022; Podell et al., 2023). Rombach et al. (2022) introduced Latent Diffusion Models (LDMs) within Stable Diffusion, using latent space for high-resolution image generation. Podell et al. (2023) extended this concept in Stable Diffusion XL (SDXL) to even higher resolutions ($1024 \times 1024$) with larger latent spaces, VAE, and U-net. Zhang & Agrawala (2023) enhances these models' capabilities by enabling the generation of controllable images conditioned on various input types. Notably, recent developments in 3D-aware 2D diffusion models, including Zero123 (Liu et al., 2023a), MVDream (Shi et al., 2023) and Sync-Dreamer (Liu et al., 2023b), have emerged. These models, also falling under the category of LDMs, can be employed to generate 3D shapes and textures by leveraging the SDS loss (Poole et al., 2023) or even reconstruction techniques (Mildenhall et al., 2021; Wang et al., 2021).

**3D shape and texture generation using 2D diffusion**  The recent method TEXTure (Yu et al., 2023) and Text2Tex (Chen et al., 2023a) can apply textures to 3D meshes using pre-trained text-to-image diffusion models, but they do not improve the mesh's shape. For the ***text-to-3D task***, DreamFusion (Poole et al., 2023) introduced the SDS loss for generating 3D objects with 2D diffusion models. Magic3D (Lin et al., 2022) extended this approach by adding a mesh optimization stage based on Stable Diffusion within the SDS loss. Subsequent works have focused on aspects like speed (Metzer et al., 2022), 3D consistency (Seo et al., 2023; Shi et al., 2023), material properties (Chen et al., 2023b), editing capabilities (Li et al., 2023), generation quality (Tsalicoglou et al., 2023; Huang et al., 2023b; Wu et al., 2023), SDS modifications (Wang et al., 2023; Zhu & Zhuang, 2023), and avatar generation (Cao et al., 2023; Huang et al., 2023a; Liao et al., 2023; Kolotouros et al., 2023). All of these works employ an SDS-like loss with Stable Diffusion. In the ***image-to-3D context***, various approaches have been explored, including those using Stable Diffusion (Melas-Kyriazi et al., 2023; Tang et al., 2023), entirely new model training (Liu et al., 2023a), and combinations of these techniques (Qian et al., 2023). Regardless of the specific approach chosen, they all rely on LDM-based SDS loss to generate 3D representations.

**Gradient clipping/normalizing techniques**  Gradient clipping and normalization techniques have proven valuable in the training of neural networks (Mikolov, 2012; Brock et al., 2021). Theoretical studies (Zhang et al., 2019; 2020; Koloskova et al., 2023) have extensively analyzed these methods. In contrast to previous ***parameter-wise*** strategies, our focus lies on the gradients of a model-rendered image. Furthermore, we introduce specifically crafted ***pixel-wise*** operations within the framework of SDS-based 3D generation. While a recent investigation by Hong et al. (2023) delves into gradient issues in 3D generation, it overlooks the impact of VAE in LDMs. In summary, we address gradient-related challenges in contemporary LDMs and crucially propose a pipeline-agnostic method for enhancing 3D generation.

## 3 BACKGROUND

### 3.1 SCORE DISTILLATION SAMPLING (SDS)

The concept of SDS, first introduced by DreamFusion (Poole et al., 2023), has transformed text-to-3D generation by obviating the requirement for text-3D pairs. SDS comprises two core elements: a 3D model and a pre-trained 2D text-to-image diffusion model. The 3D model leverages a differentiable function $x = g(\theta)$ to render images, with $\theta$ representing the 3D volume.

DreamFusion leverages SDS to synchronize 3D rendering with 2D conditioned generation, as manifested in the gradient calculation:

$$\nabla_\theta \mathcal{L}_{SDS}(\phi, g(\theta)) = \mathbb{E}_{t,\epsilon}\left[ w(t)\left(\epsilon_\phi(x_t; y, t) - \epsilon\right)\frac{\partial x}{\partial \theta}\right]. \tag{1}$$

In DreamFusion, the 2D diffusion model operates at a resolution of $64 \times 64$ pixels. To enhance quality, Magic3D (Lin et al., 2022) incorporates a 2D Latent Diffusion Models (LDM) (Rombach et al., 2022). This integration effectively boosts the resolution to $512 \times 512$ pixels, leading to an improved level of detail in the generated content.

It's important to highlight that the introduction of LDM has a subtle impact on the SDS gradient. This adjustment entails the incorporation of the gradient from the newly introduced VAE encoder, thereby contributing to an overall improvement in texture quality:

$$\nabla_\theta \mathcal{L}_{LDM-SDS}(\phi, g(\theta)) = \mathbb{E}_{t,\epsilon}\left[w(t)\left(\epsilon_\phi(z_t; y, t) - \epsilon\right)\frac{\partial z}{\partial x}\frac{\partial x}{\partial \theta}\right]. \tag{2}$$

As illustrated in the leftmost columns of Figure 2, results achieved in the latent space exhibit superior quality, highlighting a potential issue with $\partial z/\partial x$ that may impede optimization. However, due to the limited resolution of the latent space, 3D results remain unsatisfactory. Consequently, it is imperative to explore solutions to address this challenge.

## 3.2 PARAMETER-WISE NORMALIZED GRADIENT DESCENT AND GRADIENT CLIPPING

To prevent gradient explosion during training, two common strategies are typically used: Normalized Gradient Descent (NGD) and Gradient Clipping (GC) (Mikolov, 2012). These approaches both employ a threshold value denoted as $c > 0$ to a stochastic gradient, but they vary in their implementation. For a stochastic gradient $\boldsymbol{g}_t = \partial f/\partial \theta_t$, where $\theta_t$ represents a parameter, parameter-wise NGD can be expressed as

$$\theta_{t+1} = \theta_t - \eta_n \text{normalize}(\boldsymbol{g}_t), \quad \text{where normalize}(\boldsymbol{g}) := \frac{c\boldsymbol{g}}{\|\boldsymbol{g}\| + c} \tag{3}$$

where $\eta_n > 0$ denotes the learning rate. In summary, when the gradient norm $\|\boldsymbol{g}_t\|$ exceeds the threshold $c$, NGD constrains it to around $c$. However, when $\|\boldsymbol{g}_t\|$ is below $c$, NGD retains a fraction of it. A limitation of NGD becomes evident when the gradient approaches the threshold $c$.

Gradient clipping comes in two primary variants: clipping-by-value and clipping-by-norm.

**Gradient clipping-by-value** involves truncating the components of the gradient vector $\boldsymbol{g}_t$ if they surpass a predefined threshold. However, this method has a drawback, as it modifies the vector gradient's direction. This alteration in direction can influence the convergence behavior of the optimization algorithm, potentially resulting in slower or less stable training.

**Gradient clipping-by-norm** is performed in the following stochastic gradient descent iteration:

$$\theta_{t+1} = \theta_t - \eta_c \text{clip}(\boldsymbol{g}_t), \quad \text{where clip}(\boldsymbol{g}) := \min\left(\|\boldsymbol{g}\|, c\right)\frac{\boldsymbol{g}}{\|\boldsymbol{g}\|} = \min\left(\|\boldsymbol{g}\|, c\right)\boldsymbol{u}, \tag{4}$$

where $\eta_c > 0$ denotes the learning rate, $\|\boldsymbol{g}\|$ represents the norm of the gradient vector and $\boldsymbol{u}$ stands for the unit vector. By applying this operation, we ensure that the magnitude of the gradient remains below the defined threshold $c$. Notably, it also preserves the gradient's direction, offering a solution to the issues associated with the alternative method of clipping-by-value.

## 4 METHOD

To mitigate the uncontrolled term $\partial z/\partial x$, there are two viable approaches. First, during the optimization of the Variational Autoencoder (VAE), the term $\partial z/\partial x$ can be regulated. Alternatively, control over the term $\partial z/\partial x$ can be exercised during the Score Distillation Sampling (SDS) procedure. These strategies provide practical solutions to tame the erratic gradient, enhancing the stability and controllability of model training.

## 4.1 VAE OPTIMIZATION REGULATION

Managing gradient control in VAE optimization can be difficult, particularly when it's impractical to retrain both the VAE and its linked 2D diffusion model. In such cases, an alternative approach, inspired by Latent-NeRF (Metzer et al., 2022), is to train a linear layer that maps RGB pixels to latent variables. We assume a linear relationship between RGB pixels and latent variables, which allows for explicit gradient control. This control is achieved by applying L2-norm constraints to the projection matrix's norm during the training process.

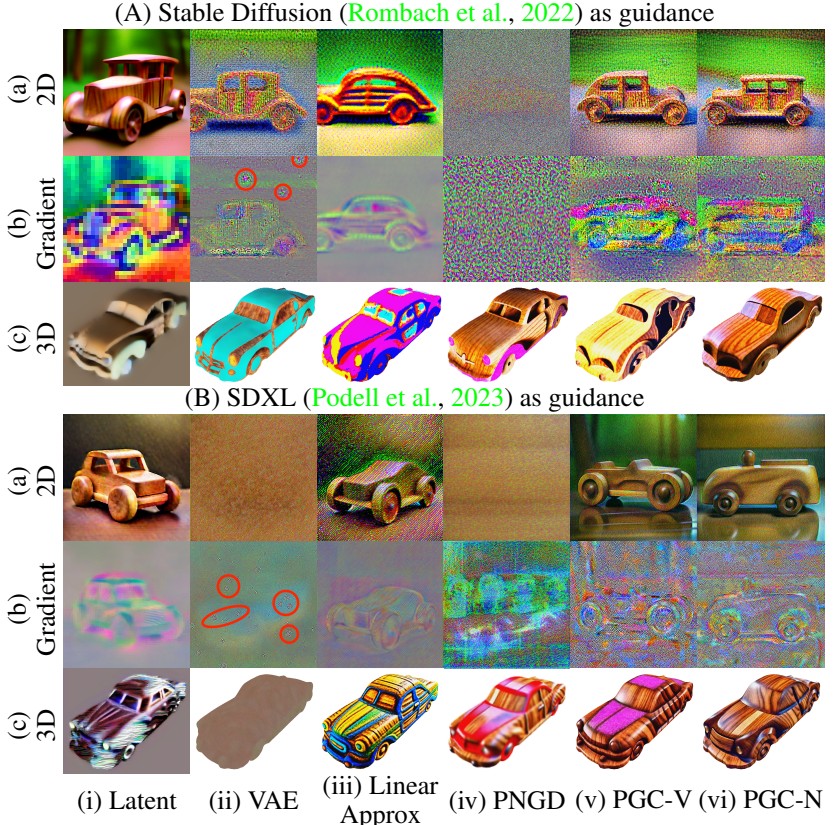

Figure 2: **Visualization of 2D/3D results and typical gradients guided by different LDMs. (A)** Stable Diffusion 2.1-base (Rombach et al., 2022) as guidance. **(B)** SDXL (Podell et al., 2023) as guidance. The text prompt is *a wooden car*. For each case, we visualize (a) directly optimizing a 2D image using SDS loss, alongside (b) the corresponding gradients; (c) optimizing a texture field (Chen et al., 2023b) based on a fixed mesh of car. We compare six gradient propagation methods: (i) Backpropagation of latent gradients, (ii) VAE gradients, (iii) linear approximated VAE gradients, (iv) normalized VAE gradients, (v) our proposed PGC VAE gradients by value and (vi) by norm. ◯ highlights gradient noise.

To elaborate, when dealing with an RGB pixel vector $x \in \mathbb{R}^3$ and a latent variable vector $y \in \mathbb{R}^4$, we establish the relationship as follows:

$$y = Ax + b, \tag{5}$$

where $A \in \mathbb{R}^{4 \times 3}$ and $b \in \mathbb{R}^4$ serve as analogs to the VAE parameters.

For evaluation, we use ridge regression methods with the COCO dataset (Lin et al., 2014) to determine the optimal configuration. For optimizing SDS, we approximate the term $\partial z / \partial x$ using the transposed linear matrix $A^\top$. This matrix is regulated through ridge regression, enabling controlled gradient behavior. Nevertheless, as illustrated in the appendix, this attempt to approximate the VAE with a linear projection falls short. This linear approximation cannot adequately capture fine texture details, thus compromising the preservation of crucial texture-related gradients.

## 4.2 Score Distillation Sampling process regulation

As previously discussed in Section 3.2, parameter-wise gradient regularization techniques are commonly employed in neural network training. Additionally, we observe that regulating gradients at the pixel level plays a crucial role in managing the overall gradient during the SDS process.

Traditionally, the training objective for 3D models is defined by 2D pixel residual, given by:

$$\mathcal{L}_{3D}(\theta, x) = \mathbb{E}[\|x - \hat{x}\|_2^2], \tag{6}$$

where $x$ denotes the rendered image, while $\hat{x}$ corresponds to the ground truth image. The objective is to minimize the disparity between the rendered image and the ground truth image. Consequently, the update rule for the 3D model can be expressed as follows:

$$\theta_{t+1} = \theta_t - \eta \frac{\partial \mathcal{L}_{3D}}{\partial \theta_t} = \theta_t - \eta \frac{\partial \mathcal{L}_{3D}}{\partial x} \frac{\partial x}{\partial \theta_t} = \theta_t - 2\eta \mathbb{E} \left[ (x - \hat{x}) \frac{\partial x}{\partial \theta_t} \right]. \quad (7)$$

For SDS on the latent variable, the gradient update for the 3D model can be simplified as:

$$\theta_{t+1} = \theta_t - \eta \mathbb{E}_{t,\epsilon} \left[ w(t) \left( \epsilon_\phi(z_t; y, t) - \epsilon \right) \frac{\partial z}{\partial x} \frac{\partial x}{\partial \theta} \right]$$
$$= \theta_t - \eta' \mathbb{E}_{t,\epsilon} \left[ \mathbb{E}_t \left[ x_t - x_{t-1} \right] \frac{\partial x}{\partial \theta} \right], \quad (8)$$

In this context, we employ the expectation of pixel residuals, denoted as $\mathbb{E}_t \left[ x_t - x_{t-1} \right]$, as a substitute for $w(t) \left( \epsilon_\phi(z_t; y, t) - \epsilon \right) \frac{\partial z}{\partial x}$.

When we compare the equation 7 and equation 8, we observe that the difference between $x$ and $\hat{x}$ remains strictly constrained within the interval of [-1, 1] due to RGB restrictions. This constraint plays a crucial role in stabilizing the training process for the 3D model. However, in the case of SDS with stochastic elements, the expectation of the 2D pixel residual, denoted as $\mathbb{E}_t \left[ x_t - x_{t-1} \right]$, is implicitly represented through the stochastic gradient $w(t) \left( \epsilon_\phi(z_t; y, t) - \epsilon \right) \frac{\partial z}{\partial x}$ without such inherent regulation. Therefore, we introduce two novel techniques: Pixel-wise Normalized Gradient Descent (PNGD) and Gradient Clipping (PGC).

### 4.3 Pixel-wise normalized gradient descent (PNGD)

PNGD incorporates a normalized gradient to regulate the change in variable $x_t - x_{t-1}$, as defined:

$$\frac{c}{\| \mathbb{E}_t \left[ x_t - x_{t-1} \right] \| + c} \mathbb{E}_t \left[ x_t - x_{t-1} \right] \quad (9)$$

PNGD effectively mitigates this issue by scaling down $\mathbb{E}_t \left[ x_t - x_{t-1} \right]$ when the gradient is exceptionally large, while preserving it when it's sufficiently small.

However, PNGD's primary limitation becomes most evident when the gradient closely approaches the threshold $c$, especially in scenarios where texture-related information is concentrated near this threshold. In such cases, the gradient norm is significantly suppressed, approaching the threshold $c$, potentially resulting in the loss of crucial texture-related details.

### 4.4 Pixel-wise gradient clipping (PGC)

To overcome PNGD's limitation, we introduce clipped pixel-wise gradients to restrict the divergence between $x_t$ and $x_{t-1}$. According to Section 3.2, the Pixel-wise Gradient Clipping (PGC) method offers two variants: Pixel-wise Gradient Clipping-by-value (PGC-V) and Pixel-wise Gradient Clipping-by-norm (PGC-N).

**PGC-V** involves directly capping the value of $\mathbb{E}[x_t - x_{t-1}]$ when it exceeds the threshold $c$. However, this adjustment affects the direction of the pixel-wise gradient, leading to a change in the correct 2D pixel residual direction. Consequently, this alteration can have a detrimental impact on the learning of real-world textures, as illustrated in the fifth column of Figure 2.

**PGC-N** can be derived from equation 4 and is expressed as follows:

$$\min \left( \| \mathbb{E} \left[ x_t - x_{t-1} \right] \|, c \right) \frac{\mathbb{E} \left[ x_t - x_{t-1} \right]}{\| \mathbb{E} \left[ x_t - x_{t-1} \right] \|} = \min \left( \| \mathbb{E} \left[ x_t - x_{t-1} \right] \|, c \right) \boldsymbol{u}_t, \quad (10)$$

where $\boldsymbol{u}_t$ stands for the unit vector.

PGC offers an advantage in managing the "zero measure" of the set created by noisy pixel-wise gradients. It achieves this by filtering out noisy gradients with negligible information while retaining those containing valuable texture information. To illustrate the effectiveness of PGC, we establish the following assumption.

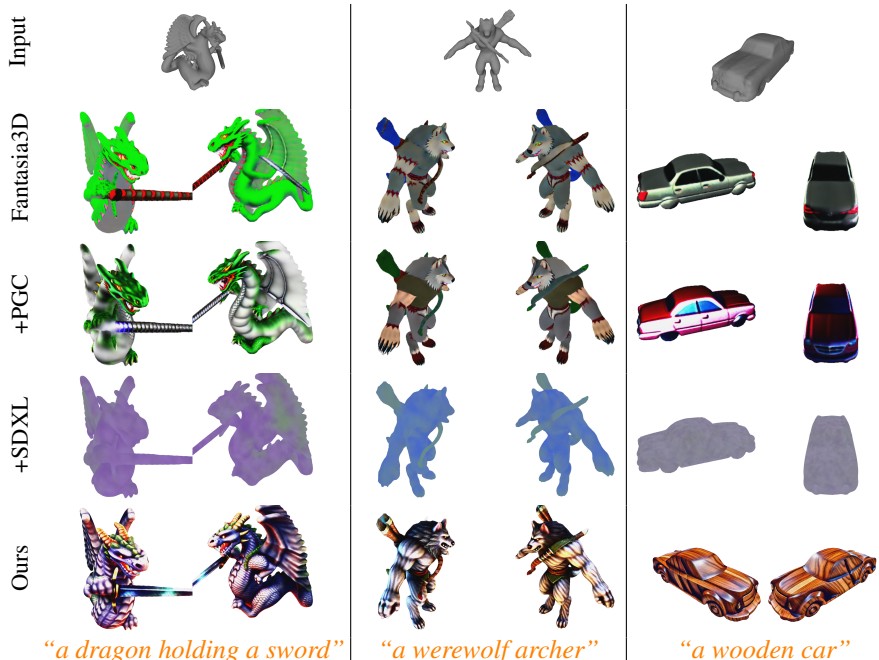

Figure 3: **Comparison with baselines.** With the meshes fixed, we compare 4 methods: Fantasia3D (Chen et al., 2023b), Fantasia3D+SDXL (Podell et al., 2023), Fantasia3D+PGC and Fantasia3D+SDXL+PGC (Ours).

### 4.4.1 NOISE ASSUMPTION

There is a concern about whether gradient clipping could lead to excessive texture detail loss, as observed in cases like linear approximation and PNGD. As shown in Figure 2's second column, we have noticed that noisy or out-of-boundary pixel-wise gradients are mainly limited to isolated points. In mathematical terms, we can assume that the gradient within the boundary is almost everywhere, with the region of being out-of-boundary having zero measure. This corresponds to the uniform boundness assumption discussed in Kim et al. (2022), which asserts that the stochastic noise in the norm of the 2D pixel residual $x_t - x_{t-1}$ is uniformly bounded by $\sigma$ for all time steps $t$: $\Pr\left[\|x_t - x_{t-1}\| \leq \sigma\right] = 1$. Furthermore, the bounded variance can be derived as $\mathbb{E}\left[\|x_t - x_{t-1}\|\right] \leq \sigma$. Now, applying Jensen Inequality to equation 10 by the convexity of L2 norm, we have:

$$\min\left(\|\mathbb{E}\left[x_t - x_{t-1}\right]\|, c\right) \leq \min\left(\mathbb{E}\left[\|x_t - x_{t-1}\|\right], c\right) \leq \min\left(\sigma, c\right) \tag{11}$$

Hence, by choosing a suitable threshold, denoted as $c \approx \sigma$, we can constrain the clipped gradient norm to remain roughly within the range of the 2D pixel residual norm, represented by $\sigma$. This approach is essential as it enables the preservation of pixel-wise gradient information without excessive truncation. This preservation effectively retains texture detail while ensuring noise remains within acceptable limits.

### 4.5 CONTROLLABLE LATENT GRADIENTS

Improper gradients in the latent space can result in failure scenarios. This can manifest as a noticeable misalignment between the visualized gradients and the object outlines in the rendered images, causing a texture mismatch with the mesh. To mitigate this issue, we propose incorporating shape information into U-nets. Leveraging the provided mesh, we apply a depth and/or normal controlnet (Zhang & Agrawala, 2023), substantially enhancing the overall success rate.

## 5 EXPERIMENTS

### 5.1 IMPLEMENTATION DETAILS

For all the experiments, we adopt the uniform setting without any hyperparameter tuning. Specifically, we optimize the same texture and/or signed distance function (SDF) fields as Chen et al.

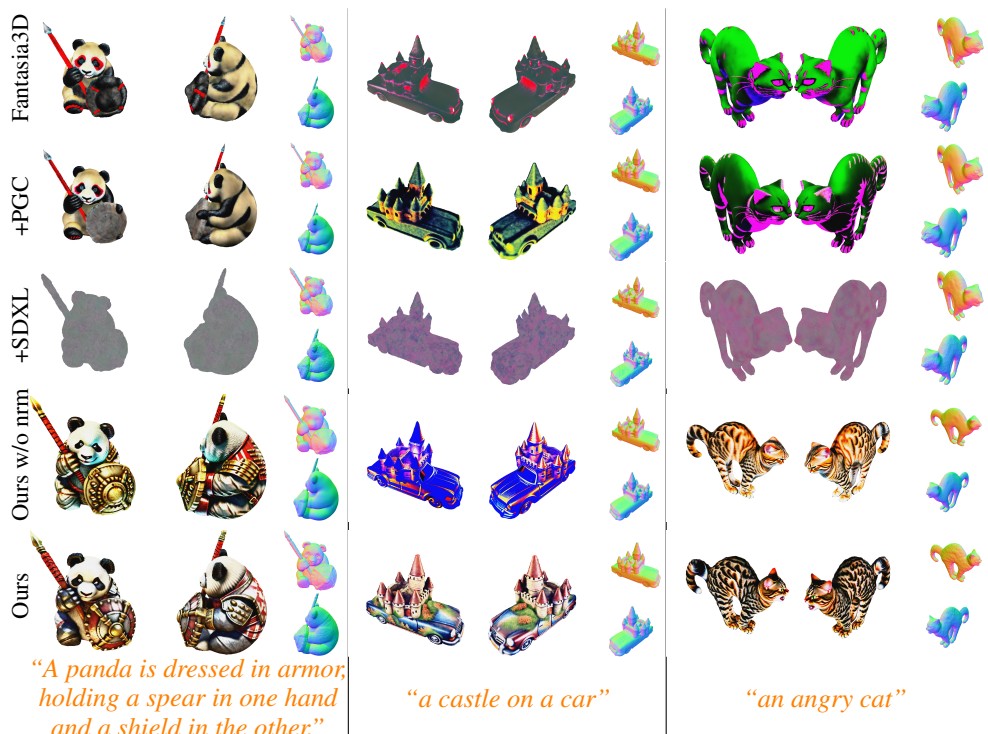

Figure 4: **Comparison of using normal-SDS jointly with RGB-SDS.** We compare 5 methods: Fantasia3D (Chen et al., 2023b), Fantasia3D+SDXL (Podell et al., 2023), Fantasia3D+PGC, Fantasia3D+SDXL+PGC (Ours) and Fantasia3D+SDXL+PGC w/o normal-SDS (Ours w/o nrm).

(2023b) for 1200 iterations on two A6000 GPUs with batch size 4 by using Adam optimizer without weight decay. The learning rates are set to constant $1 \times 10^{-3}$ for texture field and $1 \times 10^{-5}$ for SDF field. For the sampling, we set the initial mesh normalized in $[-0.8, 0.8]^3$, focal range $[0.7, 1.35]$, radius range $[2.0, 2.5]$, elevation range $[-10°, 45°]$ and azimuth angle range $[0°, 360°]$. In SDS, we set CFG 100, $t \sim U(0.02, 0.5)$ and $w(t) = \sigma_t^2$. In PGC, we use PGC-N for PGC as default and set the threshold $c = 0.1$. The clipping threshold is studied in Section A.3.

## 5.2 PGC ON MESH OPTIMIZATION

As Stable Diffusion (Rombach et al., 2022) and Stable Diffusion XL (SDXL) (Podell et al., 2023) demonstrate notable capabilities in handling high-resolution images, our primary focus lies in evaluating PGC's performance within the context of mesh optimization, with a specific emphasis on texture details. To conduct comprehensive comparisons, we employ numerous mesh-prompt pairs to optimize both texture fields and/or SDF fields. Our experimental framework establishes the Fantasia3D's appearance stage, utilizing Stable Diffusion-1.5 with depth-controlnet for albedo rendering, as our baseline reference. Subsequently, we conduct a series of methods, including Fantasia3D+PGC, Fantasia3D+SDXL (replace Stable Diffusion), and Fantasia3D+SDXL+PGC.

In the first setting where the meshes remain unchanged, the outcomes of these comparisons are presented in Figure 3. It is noteworthy that PGC consistently enhances texture details when contrasted with the baseline. Notably, the direct replacement of Stable Diffusion with SDXL results in a consistent failure; however, the integration of PGC effectively activates SDXL's capabilities, yielding a textured mesh of exceptional quality.

In the second setting, we allow for alterations in mesh shape through the incorporation of normal-SDS loss which replaces RGB image with normal image as the input of diffusion model, albeit at the expense of doubling the computation time. The results of these experiments are presented in Figure 4. Similar to the first setting, we observe a consistent enhancement in texture quality by using PGC. Furthermore, in terms of shape details, the utilization of normal-SDS loss yields significantly more intricate facial features in animals. Interestingly, we find that even if the change of shape is

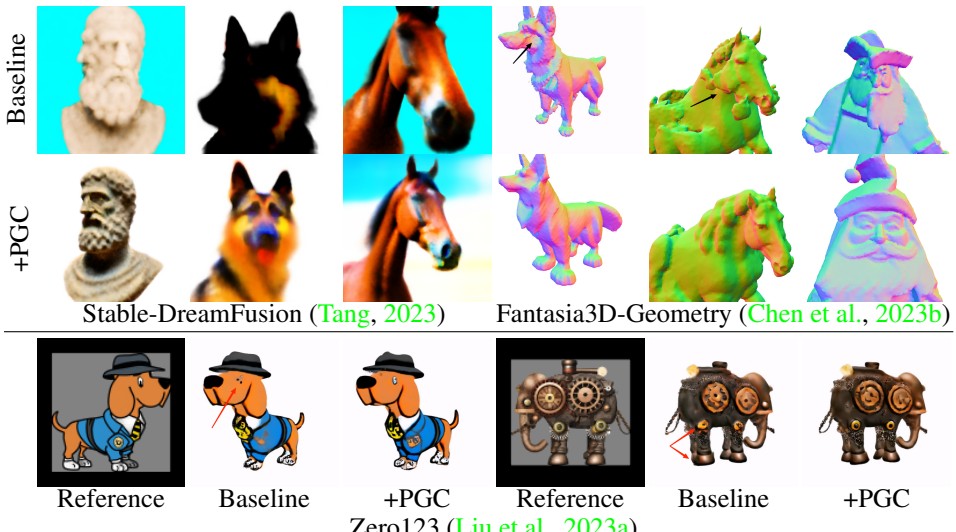

Figure 5: PGC can benefit various pipelines, including Stable-Dreamfusion (Tang, 2023) , Fantasia3D (Chen et al., 2023b) geometry stage and Zero123 (Liu et al., 2023a).

not particularly significant, minor perturbation of the input points coordinates of texture fields can enhance the robustness of optimization, resulting in more realistic texture.

## 5.3 PGC BENEFITS VARIOUS PIPELINES

We also test PGC in various pipelines using LDM: Stable-DreamFusion (Tang, 2023) with Stable Diffusion 2.1-base, Fantasia3D (Chen et al., 2023b) geometry stage with Stable Diffusion 2.1-base and Zero123-SDS (Liu et al., 2023a). These three pipelines cover a wide range of SDS applications including both text-to-3d and image-to-3d tasks. As depicted in Figure 5, within the Stable-DreamFusion pipeline, PGC demonstrates notable improvements in generation details and success rates. In the case of Fantasia3D, PGC serves to stabilize the optimization process and mitigate the occurrence of small mesh fragments. Conversely, in the Zero123 pipeline, the impact of PGC on texture enhancement remains modest, primarily due to the lower resolution constraint at 256. However, it is reasonable to anticipate that PGC may exhibit more pronounced effectiveness in scenarios involving larger multi-view diffusion models, should such models become available in the future.

## 5.4 USER STUDY

We also conducted user study to evaluate our methods quantitatively. We put 12 textured meshes generated by 4 methods described in Section 5.2 on website so that users are able to conveniently rotate and scale 3D models for observation online and finally pick the preferred one. Among 15 feedback with 180 picks, ours received 84.44% preference while Fantasia3D w/ and w/o PGC only received 10.56% and 5% preference, respectively. Since Fantasia3D+SDXL w/o PGC does not generate any meaningful texture, no one picks this method. The results show that our proposed PGC greatly improves generation quality. More results can be found in supplementary materials.

## 6 CONCLUSION

In our research, we have identified a critical and widespread problem when optimizing high-resolution 3D models: the uncontrolled behavior of pixel-wise gradients during the backpropagation of the VAE encoder's gradient. To tackle this issue, we propose an efficient and effective solution called Pixel-wise Gradient Clipping (PGC). This technique builds upon traditional gradient clipping but tailors it to regulate the magnitudes of pixel-wise gradients while preserving crucial texture information. Theoretical analysis confirms that the implementation of PGC effectively bounds the norm of pixel-wise gradients to the expectation of the 2D pixel residual. Our extensive experiments further validate the versatility of PGC as a general plug-in, consistently delivering benefits to existing SDS and LDM-based 3D generative models. These improvements translate into significant enhancements in the realm of high-resolution 3D texture synthesis.

**Acknowledgments** This work was supported in part by STI2030-Major Projects (Grant No. 2021ZD0200204), National Natural Science Foundation of China (Grant No. 62106050 and 62376060), Natural Science Foundation of Shanghai (Grant No. 22ZR1407500), USyd-Fudan BISA Flagship Research Program and and Lingang Laboratory (Grant No. LG-QS-202202-07).

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

## A    APPENDIX

### A.1    POTENTIAL RISKS AND SOCIAL IMPACTS

Every method that learns from data carries the risk of introducing biases. Our 3D generation model is based on the text-to-image models that are pre-trained on the image and text data from the Internet. Work that bases itself on our method should carefully consider the consequences of any potential underlying biases.

### A.2    LINEAR APPROXIMATION FOR VAE

We provide linear approximation between image pixel $x \in \mathbb{R}^3$ and latent pixel $z \in \mathbb{R}^4$: $x = A_0 z + b_0$ and $z = A_1 x + b_1$, where

$$A_0 = \begin{bmatrix} -0.5537 & 1.8844 & 2.1757 \\ -3.4900 & 1.7472 & 1.6805 \\ 0.6894 & 3.2756 & -3.4658 \\ -2.4909 & 1.3309 & -0.1115 \end{bmatrix} \quad b_0 = \begin{bmatrix} -1.6590 \\ 0.3810 \\ -0.3939 \\ 0.7896 \end{bmatrix}$$

$$A_1 = \begin{bmatrix} 0.1956 & -0.0910 & 0.0462 & -0.1521 \\ 0.2125 & -0.0206 & 0.0401 & -0.1215 \\ 0.2208 & 0.0047 & -0.0028 & -0.1083 \end{bmatrix} \quad b_1 = \begin{bmatrix} 0.5573 \\ 0.5105 \\ 0.4635 \end{bmatrix}$$

Figure 6 visualizes samples of fitted image-latent pair.

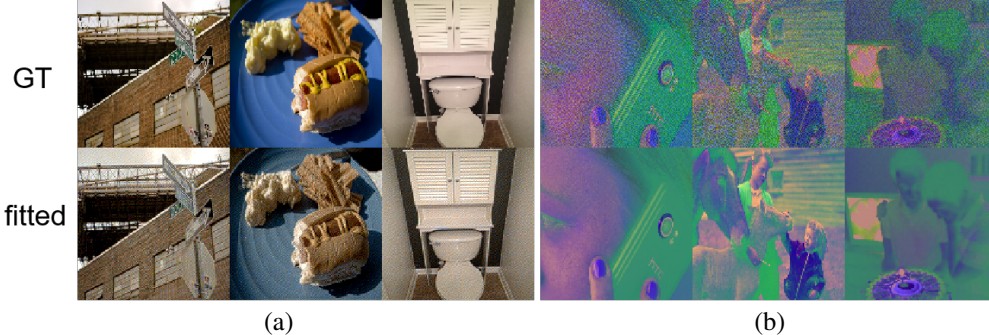

(a)                     (b)

Figure 6: Samples of linear approximation. (a) The results of VAE decoder approximation. (b) The results of VAE encoder approximation. For each case, top is the ground-truth and bottom is the fitted result.

### A.3    ABLATION ON CLIPPING THRESHOLD

Figure 7 shows the texture quality is robust to different clipping thresholds.

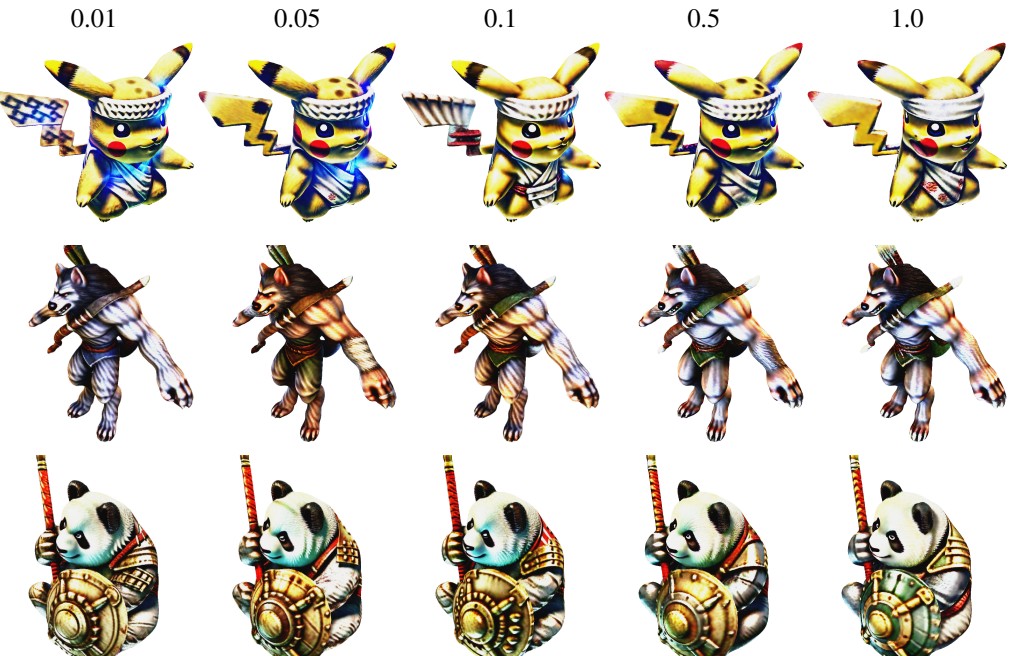

Figure 7: **Ablation on clipping threshold.** The thresholds are chosen from {0.01, 0.05, 0.1, 0.5, 1.0}.

