# Supplementary materials for "Enhancing High-Resolution 3D Generation through Pixel-wise Gradient Clipping"

## 1 More results

### 1.1 More our results

**Editing results.** With the same base mesh, we provide texture editing results based on different input prompts in Figure 1.

**Text-to-texture results.** In Figure 2, we present meshes with high-quality textures.

### 1.2 More ablations

**Depth controlnet** In Figure 3, we provide two cases to illustrate the effect of depth controlnet in mesh optimization Zhang & Agrawala (2023). In the left case, both settings produce high quality textures. In the right case, the generated texture at the absence of depth controlnet is not aligned with the mesh geometry. In general, only about 40% cases succeed without depth controlnet and the rate increases to 60%-70% by incorporating the depth controlnet.

**Parameter-wise NGD and GC** Parameter-wise normalized gradient descent (NGD) and gradient clipping (GC) will allow the extremely large and unbalanced gradients of the rendered image to be back-propagated until the gradients of the earliest parameters of NeRF/DMTet. As shown in Figure 4, parameter-wise NGD and GC cannot produce as detailed results as Pixel-wise Gradient Clipping under the text-to-texture setting using SDXL.

### 1.3 More comparisons

**Pixel-wise Gradient Clipping benefits ProlificDreamer.** We use threestudio Guo et al. (2023) implementation of ProlificDreamer Wang et al. (2023) and adopt Stable Diffusion as guidance. In Figure 5, we observe that Pixel-wise Gradient Clipping stabilizes the training and avoid bad geometry and floaters.

**Pixel-wise Gradient Clipping benefits improved Fantasia3D.** As suggested in the official implementation of Fantasia3D Chen et al. (2023), alternative weighting strategy $w(t) = \frac{1}{\sigma_t^2}$ and negative prompts can help generate better texture. As shown in Figure 6, our proposed Pixel-wise Gradient Clipping also benefits improved Fantasia3D under this setting.

**Mesh optimization comparison.** We present more comparison results with Fantasia3D (Chen et al., 2023) baseline as shown in Figure 7 and Figure 8.

**Image-to-3D comparison.** Figure 9 shows one more case using Zero123 (Liu et al., 2023) SDS loss.

## References

Rui Chen, Yongwei Chen, Ningxin Jiao, and Kui Jia. Fantasia3d: Disentangling geometry and appearance for high-quality text-to-3d content creation. In *ICCV*, 2023.

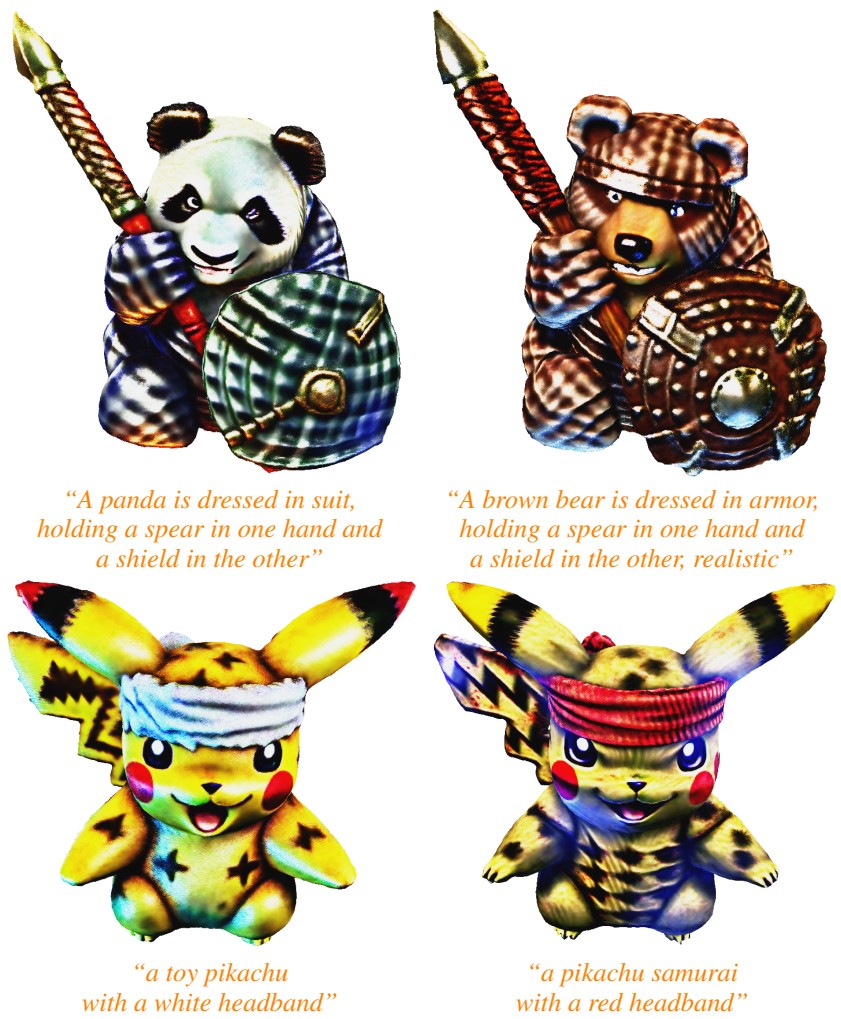

*"A panda is dressed in suit, holding a spear in one hand and a shield in the other"*

*"A brown bear is dressed in armor, holding a spear in one hand and a shield in the other, realistic"*

*"a toy pikachu with a white headband"*

*"a pikachu samurai with a red headband"*

Figure 1: Editing results based on different text prompts.

Yuan-Chen Guo, Ying-Tian Liu, Chen Wang, Zi-Xin Zou, Guan Luo, Chia-Hao Chen, Yan-Pei Cao, and Song-Hai Zhang. threestudio: A unified framework for 3d content generation. `https://github.com/threestudio-project/threestudio`, 2023.

Ruoshi Liu, Rundi Wu, Basile Van Hoorick, Pavel Tokmakov, Sergey Zakharov, and Carl Vondrick. Zero-1-to-3: Zero-shot one image to 3d object. In *ICCV*, 2023.

Dustin Podell, Zion English, Kyle Lacey, Andreas Blattmann, Tim Dockhorn, Jonas Müller, Joe Penna, and Robin Rombach. Sdxl: improving latent diffusion models for high-resolution image synthesis. *arXiv preprint*, 2023.

Zhengyi Wang, Cheng Lu, Yikai Wang, Fan Bao, Chongxuan Li, Hang Su, and Jun Zhu. Prolific-dreamer: High-fidelity and diverse text-to-3d generation with variational score distillation. *arXiv preprint*, 2023.

Lvmin Zhang and Maneesh Agrawala. Adding conditional control to text-to-image diffusion models. *arXiv preprint*, 2023.

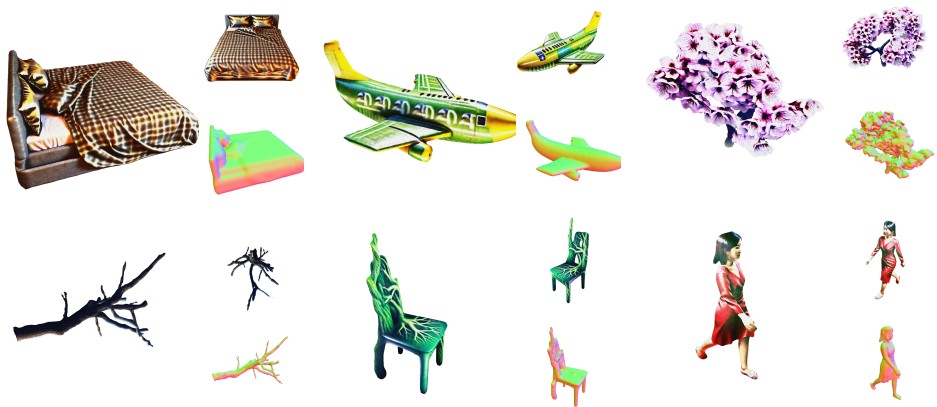

Figure 2: More our generated high-quality textures.

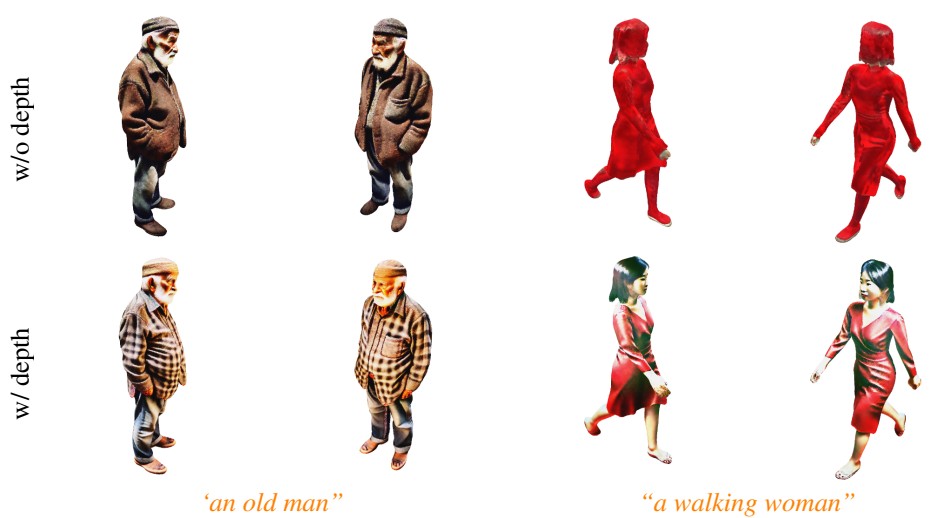

*'an old man"*      *"a walking woman"*

Figure 3: Ablation on depth controlnet. Depth controlnet can increase the success rate.

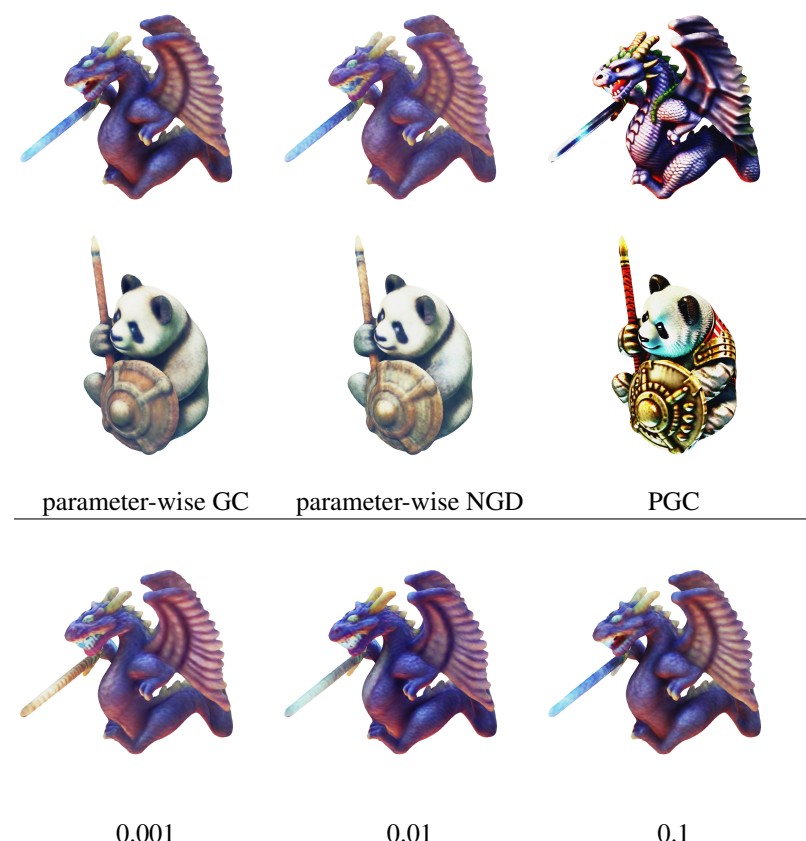

| parameter-wise GC | parameter-wise NGD | PGC |
|---|---|---|

| 0.001 | 0.01 | 0.1 |
|---|---|---|

Figure 4: Comparing our proposed Pixel-wise Gradient Clipping (PGC) with parameter-wise GC and NGD. The third row shows the ablation results on clipping value used in parameter-wise GC. It is evdient that our PGC achieves much superior results.

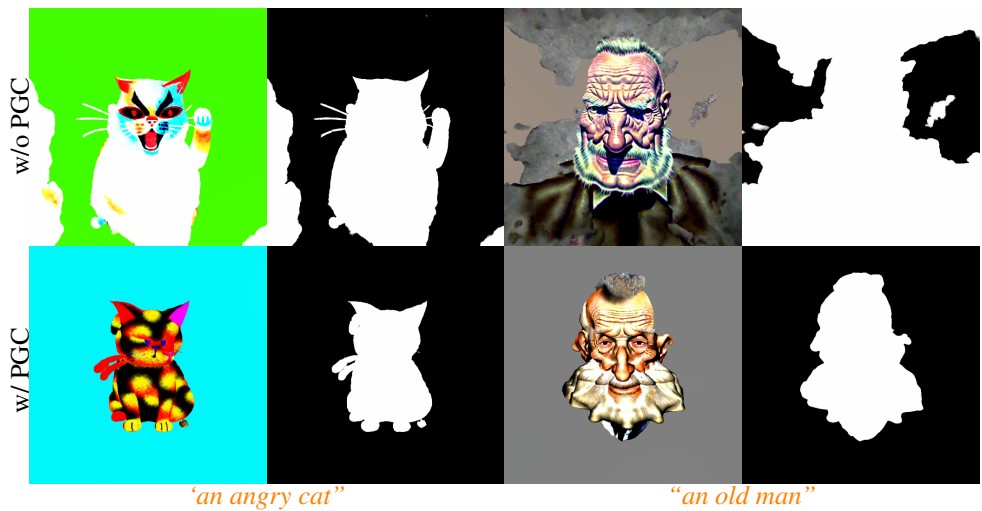

"an angry cat"    "an old man"

Figure 5: Evaluating the effect of our Pixel-wise Gradient Clipping (PGC) within Prolific-Dreamer Wang et al. (2023). We present RGB image and mask for each case. Our PGC also stabilizes the training using VSD loss, leading to favoured geometry.

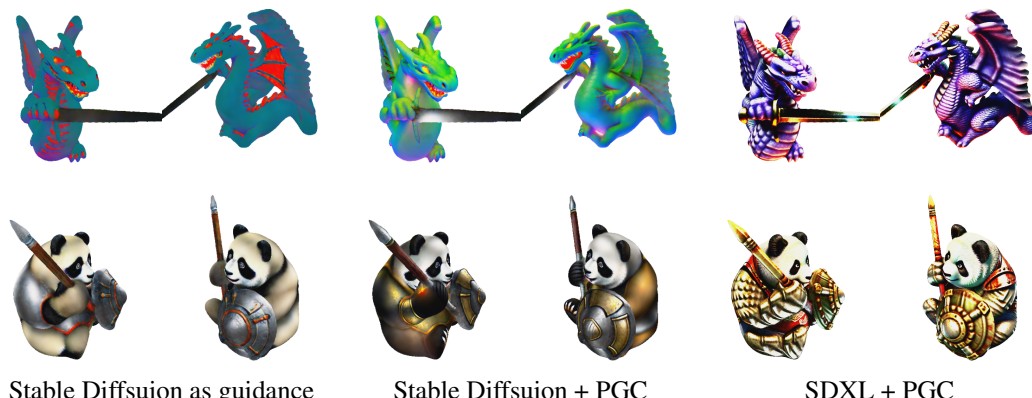

Stable Diffsuion as guidance          Stable Diffsuion + PGC          SDXL + PGC

Figure 6: Evaluating with the improved Fantasia3D Chen et al. (2023) using alternative weighting strategy and negative prompts. Our proposed Pixel-wise Gradient Clipping (PGC) improves the texture quality consistently under this setting. PGC also enables leveraging the more advanced SDXL Podell et al. (2023), producing more detailed and realistic texture compared to using Stable Diffusion.

Fantasia3D (Chen et al., 2023)                                          Ours

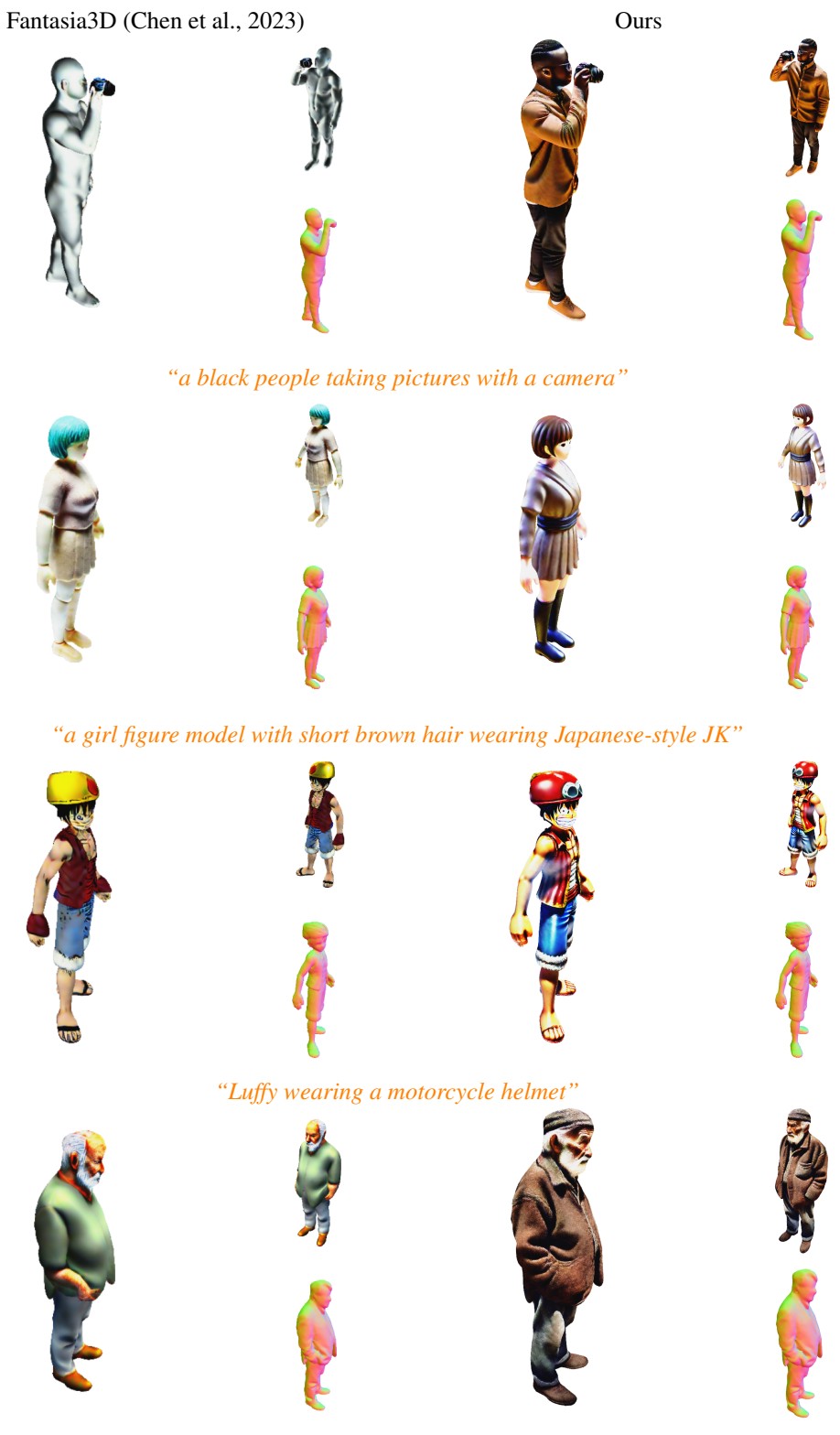

*"a black people taking pictures with a camera"*

*"a girl figure model with short brown hair wearing Japanese-style JK"*

*"Luffy wearing a motorcycle helmet"*

*"an old man"*

Figure 7: More comparison results.

Fantasia3D (Chen et al., 2023)                    Ours

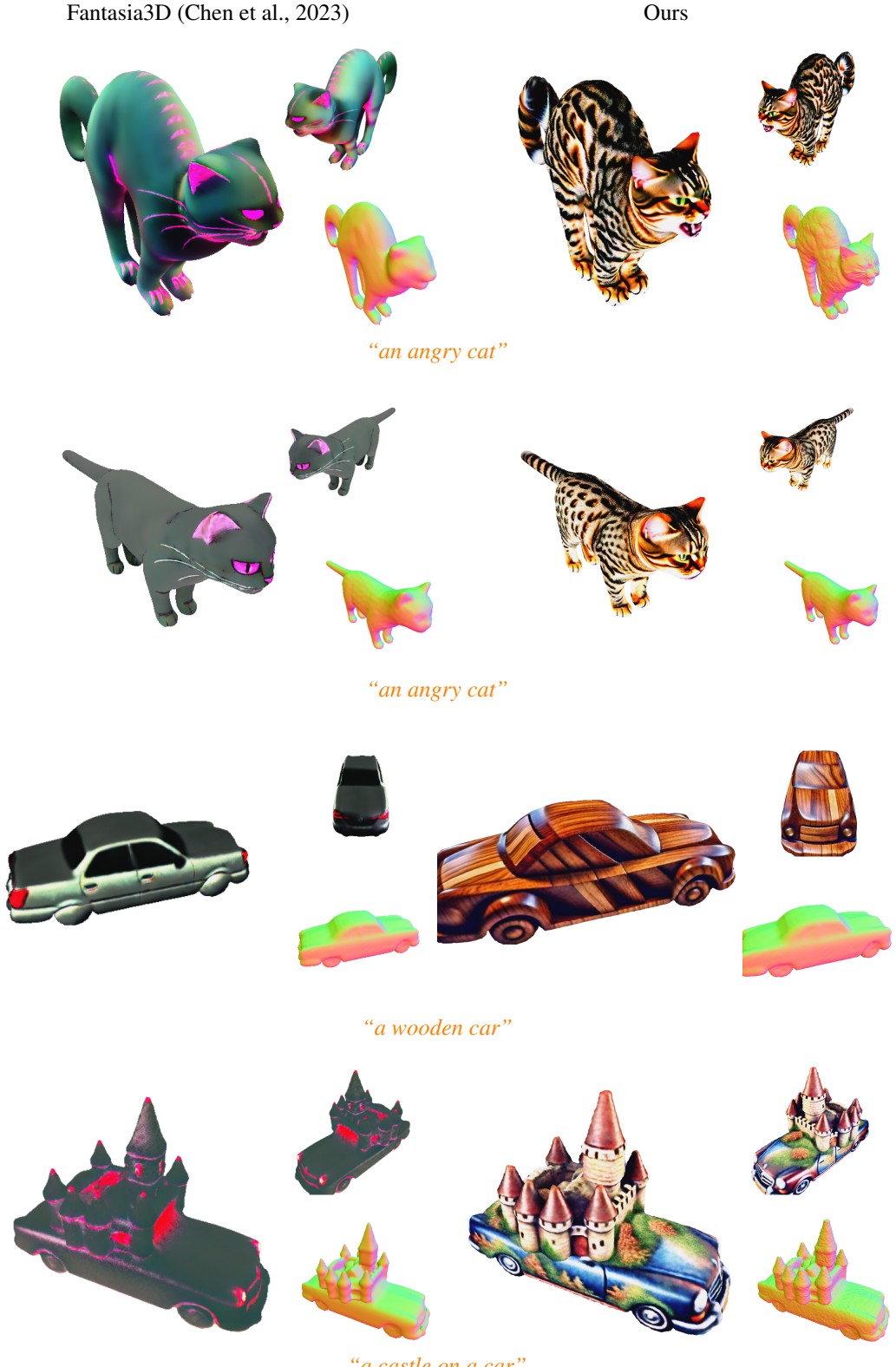

*"an angry cat"*

*"an angry cat"*

*"a wooden car"*

*"a castle on a car"*

Figure 8: More comparison results.

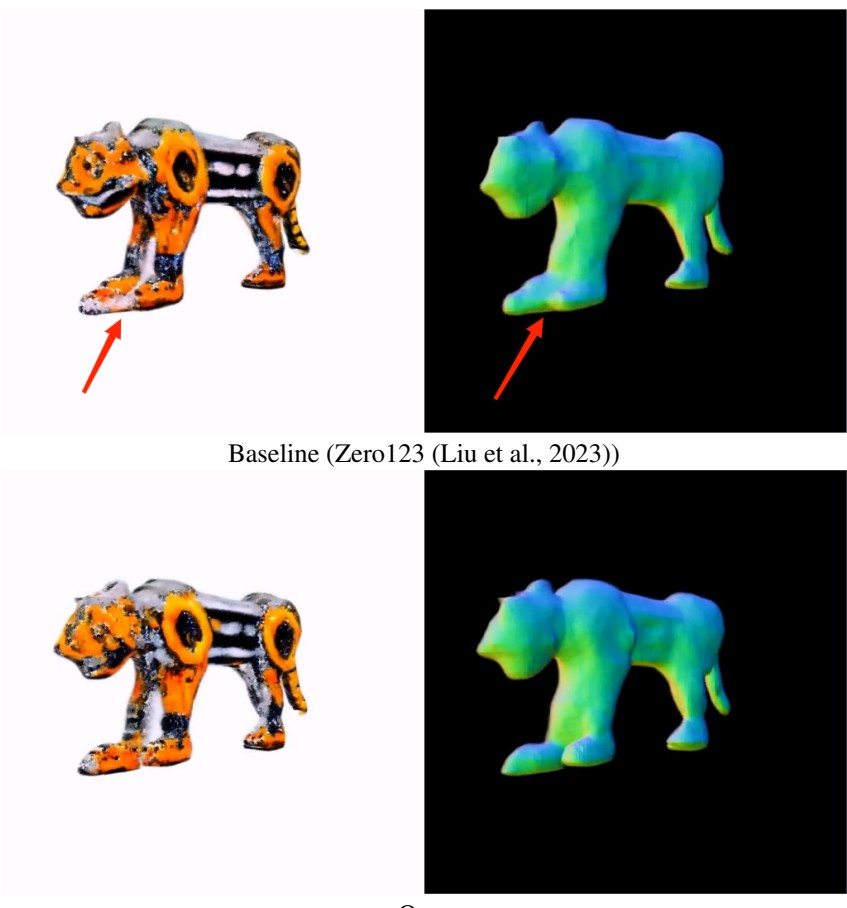

Baseline (Zero123 (Liu et al., 2023))

Ours

Figure 9: Image-to-3D comparison.