# OpenReview forum: "Enhancing High-Resolution 3D Generation through Pixel-wise Gradient Clipping"
_ICLR.cc/2024/Conference — ICLR 2024 poster_

### Official Review · Reviewer_L63f · 2023-10-18

**Soundness:** 4 excellent
**Presentation:** 3 good
**Contribution:** 3 good
**Rating:** 6
**Confidence:** 4

**Summary:**

The paper identifies a critical and generic issue in optimizing high-resolution 3D models by exploiting latent diffusion models. To address this problem, the paper analyzes the gradient propagation process and proposes a simple and effective gradient clipping technique. The paper demonstrates the proposed techniques can be used as a generic plug-in method to improve a body of works based on LDM.

**Strengths:**

The paper identifies an important problem in LDM based 3D AIGC method, and proposes a simple and effective technique to solve this problem. I like the general idea of this paper. I think it can benefit a large amount of future work in this field.

**Weaknesses:**

(1) The most important Figure in the paper, i.e. Figure 2, is unclear to me.
* Figure 2 is first referenced in Section 3.1. For me, it's very hard to understand the meaning and experiment setup of this Figure. What's the meaning of the three rows respectively? What's the task of Figure 2, is the mesh fixed while only optimizing the texture? Many important details are missed in my eyes; it makes me feel hard to follow the paper.

(2) Some other figures and descriptions are unclear to me. More necessary details should be provided.
* In Figure 6, what is the meaning of the 6 images in the right column? Is the gradient?
* Can the author provide more illustrations to support Section 4.5?

**Questions:**

(1) In Section 4.2, it says "we observe that the difference between x and \hat{x} remains strictly constrained within the interval of (0, 1) due to RGB restrictions". I think the range of the difference should be in [-1,1]. Did I miss something?

(2) Can the author provide some insights on why "this constraint plays a crucial role in stabilizing the training process for the 3D model"?

(3) Another way is to clip the gradient of the entire term, i.e. (eps_phi - eps) * dz/dx, rather than the dz/dx. Can the author comment on this alternative method?

---

> ### Author Response · Authors · 2023-11-21
>
> **Q1: Description for Figure 2 is unclear.**
>
> Thanks for pointing it out. For both Stable Diffusion and SDXL, we present images organized in 3 rows and 6 columns. Let’s explain them in detail one by one.
>
> Row1: The setting is training a 2D image using SDS loss with different gradients.
>
> Row2: The visualized gradients of the optimized image as shown in Row1.
>
> Row3: The setting is training a texture field for a fixed mesh using SDS loss with different gradients.
>
> Column1: The setting is training 2D image/3D texture field in latent space as Latent-NeRF.
>
> Column2-6: training them in image space where the VAE encoder will be incorporated in gradient backward.
>
> Column2: Using original gradient backward through VAE encoder. (VAE gradient)
>
> Column3: Using a linear approximation for VAE encoder.
>
> Column4: Using the pixel-wise normalized VAE gradient.
>
> Column5: Using the pixel-wise clipped VAE gradient based on value.
>
> Column6: Using the pixel-wise clipped VAE gradient based on norm.
>
> We have made the caption of Figure 2 more clear in the revision.
>
> **Q2: Figure 6 is unclear.**
>
> Thanks for pointing it out. Since VAE has two parts: encoder and decoder, we fit linear approximation for both parts. The left three columns are the results of decoder approximation where fitted and GT RGB images are presented.
> The right three columns are the results of encoder approximation where fitted and GT latent images are presented.
> We have made a clear description for them in the revision.
>
> **Q3: More illustrations to support Section 4.5 (depth controlnet).**
>
> We have provided two cases to illustrate the problem in Section 1.2 of the **separate supplementary file**. Depth controlnet improves the success rate of generating high quality meshes like the bottom of Figure 1. In our observation, only about 40% cases succeed without depth controlnet and the rate increases to 60%-70% by incorporating the depth controlnet. In the rest cases, while the quality is not as high as in Figure 1, our PGC could still  enhance  over the baseline.
>
> **Q4: Typo of the range (0,1).**
>
> Yes, it is a typo and should be [-1, 1]. We appreciate your careful reading and have fixed the typo in the revision.
>
> **Q5: More insights of the constraint.**
>
> Great comment. More specifically, we note that RGB values are normalized to the range of [0, 1] while the absolute value of the image gradients produced by SDS loss can vary from $0$ to $10^{5}$. If we optimize a 2D image directly using these gradients with SGD, the optimized image will quickly explode thereby leading to failure. Although sigmoid function and Adam optimizer are often used to guarantee the RGB constraints and smoother gradients (as all the experiments in the main paper), we think the extremely large and unbalanced gradients still do not make sense.
>
> **Q6: Alternative method by clipping $(\epsilon_{\phi} -\epsilon) \frac{dz}{dx}$ rather than the  $\frac{dz}{dx}$.**
>
> Apologies for this misunderstanding. In our design, we take operations on $w(t)(\epsilon_{\phi} -\epsilon) \frac{\partial z}{\partial x}$ which are the gradients of an image. The term $\frac{\partial z}{\partial x}$  is the Jocobian matrix of size $(128\times 128 \times 4) \times (1024\times 1024 \times 3)$ under SDXL setting. We have revised the relevant expressions in the paper.

---

> > ### Comment · Reviewer_L63f · 2023-11-23
> >
> > Thanks for the rebuttal. All my questions are clarified. I would keep my rating.

---

### Official Review · Reviewer_ezZ1 · 2023-10-31

**Soundness:** 2 fair
**Presentation:** 2 fair
**Contribution:** 2 fair
**Rating:** 6
**Confidence:** 4

**Summary:**

This paper aims to address gradient-related issues in typical Latent Diffusion Models used for 3D generation. This paper finds that the unregulated gradients through image models are harmful for the 3D models to capture correct and fine textures. To solve this problem, this paper proposes Pixel-wise Gradient Clipping (PGC) that clips the pixel-wise gradients by norm. Experiments show improved performance over baselines such as Fantasia3D.

**Strengths:**

1. The paper is well-written and easy to follow and understand. For example, section 3 provides enough background on score distillation sampling and gradient clipping to motivate the observations of optimization issue in existing models and the proposals on pixel-wise gradient clipping.

2. This paper targets at an important goal: improving gradient flows that 3D models obtain from 2D diffusion models. This is critical because many existing text/image-to-3D models rely on pretrained 2D diffusion models to guide the optimization of a 3D model. An effective tool can potentially benefit lots of related work.

3. The proposed PGC seem to be effective over a few baselines such as Fantasia3D+SDXL.

**Weaknesses:**

1. Insufficient ablation study
The paper reviews parameter-wise normalized Gradient Descent (NGD) and Gradient Clipping (GC) in Sec. 3.2 & 3.3 and proposes pixel-wise NGD and GC in Sec. 4.3 & 4.4. Pixel gradients are computed from parameter gradients, and bounded parameter gradients may result in bounded pixel gradients too -- an alternative of the proposed Sec. 4.3 & 4.4 to limit pixel gradient magnitudes. Therefore, parameter-wise NGD and GC would have served as good baselines for ablation study. However, the paper only showed results of pixel-wise NGD and GC but not results of parameter-wise NGD and GC or other simple ways to bound pixel gradients.

2. Insufficient experiments to show that the proposed method "benefit existing SDS and LDM-based 3D generative models"
The paper claims "PGC consistently benefit existing SDS and LDM-based 3D generative models", but experiments only compared with two baselines: Stable-DreamFusion and (SDXL variant of) Fantasia3D, which are not published state-of-the-arts and not sufficient to support the claim. For example, ProlificDreamer [Wang et al. 2023] and improved Fantasia3D (https://github.com/Gorilla-Lab-SCUT/Fantasia3D "Q7") can produce more realistic texture results than the compared baselines. Will the proposed PGC still be effective on top these two methods?

**Questions:**

1. Since this paper mostly focuses on addressing the gradient issues, I would suggest adding a paragraph in Sec. 2 to review general gradient-related techniques and their connections&differences with the proposed method. Example related papers include:
[a] Zhang et al. Why Gradient Clipping Accelerates Training: A Theoretical Justification for Adaptivity. ICLR 2020.
[b] Zhang et al. Improved analysis of clipping algorithms for non-convex optimization. NeurIPS 2020.
[c] Brock et al. High-Performance Large-Scale Image Recognition Without Normalization. ICML 2021.
[d] Koloskova et al. Revisiting Gradient Clipping: Stochastic bias and tight convergence guarantees. ICML 2023.

2. Minor suggestions:
(1) Currently the texts that refer to Fig. 2 are scattered all around. It may improve the clarity a lot by adding a few sentences in the caption to explain different columns in Fig. 2.
(2) Typo: last paragraph before Sec. 4.3: "xt-1" --> "x_{t-1}"

---

> ### Author Response · Authors · 2023-11-21
>
> **Q1: More ablations on parameter-wise NGD and GC.**
>
> First we note that when we conduct 2D experiments (i.e. just optimizing a 2D image as the first row of Figure 2), “parameter-wise” design is equal to the “pixel-wise” counterpart because a pixel itself is a parameter to be optimized. For 3D experiments, parameter-wise NGD and GC will allow the extremely large and unbalanced gradients of the rendered image to be back-propagated until the gradients of the earliest parameters of NeRF/DMTet. This causes suboptimal results.
>
> Importantly, we provide the evidence in the Section 1.2 of the **separate supplementary file** (Figures 4) that parameter-wise NGD and GC cannot produce as detailed results as our PGC under the text-to-texture setting using SDXL.
>
> **Q2: More comparisons.**
>
> Thanks. We have now provided the results of ProlificDreamer and improved Fantasia3D in the Section 1.3 of the **separate supplementary file** (Figures 5 and 6).
>
> For ProlificDreamer, we use Stable Diffusion as guidance. We observe that our PGC also stabilizes the training. We  cannot test the combination of SDXL and VSD due to the restriction of GPU memory. Overall, our method enables the unleashing of the potentials of the advanced diffusion model successfully.
>
> For improved Fantasia3D, we have already tried the alternative weighting strategy: $w(t)=\frac{1}{\sigma_t^2}$ mentioned in their readme Q7. Since threestudio and stable-dreamfusion have not implemented this strategy, we did not include this for fair comparisons in our submission. In our experiments, we find that our proposed PGC consistently enhances  texture quality.
>
> **Q3: Related work for gradient-related techniques.**
>
> Thanks. We have added extra discussion in the related work following this suggestion. We note an important difference that previous works focus on model parameters while we manipulate the images –  the intermediate variables in the forward/backward process.
>
> **Q4: Description for Figure 2 is unclear.**
>
> Thanks for pointing it out. For both Stable Diffusion and SDXL, we present images organized in 3 rows and 6 columns. We explain them in detail one by one.
>
> Row1: The setting is training a 2D image using SDS loss with different gradients.
>
> Row2: The visualized gradients of the optimized image as shown in Row1.
>
> Row3: The setting is training a texture field for a fixed mesh using SDS loss with different gradients.
>
> Column1: The setting is training 2D image/3D texture field in latent space as Latent-NeRF.
>
> Column2-6: training them in image space where the VAE encoder will be incorporated in gradient backward.
>
> Column2: Using original gradient backward through VAE encoder. (VAE gradient)
>
> Column3: Using a linear approximation for VAE encoder.
>
> Column4: Using the pixel-wise normalized VAE gradient.
>
> Column5: Using the pixel-wise clipped VAE gradient based on value.
>
> Column6: Using the pixel-wise clipped VAE gradient based on norm.
>
> We have made the caption of Figure 2 more clear in the revision.
>
> **Q5: Typo.**
>
> We appreciate your careful reading. We have fixed the typo in the revision.

---

> > ### Comment · Reviewer_ezZ1 · 2023-11-22
> >
> > I appreciate the authors' efforts in trying to address the concerns from all reviewers. I have read through all reviewer comments as well as the rebuttals and revised drafts from the authors. What is the threshold set for the parameter-wise NGD/GC? I thought if the threshold is small enough, "the extremely large and unbalanced gradients of the rendered image" will not happen.

---

> ### Author Response · Authors · 2023-11-23
>
> Thanks. We set the threshold as 0.1 for all parameters. We have updated Figure 4 in the supplementary file by adding the ablation on clipping value used in parameter-wise GC. The clipping value ranges from $10^{-3}$ to $10^{-1}$. Parameter-wise clipping led to the generation of blurry textures in all settings. This approach appears inadequate in addressing **unbalanced** gradients as the earliest gradients of model parameters remain correlated with high-magnitude noise in image gradients, potentially overwhelming the lower-magnitude gradients during the backward process. Such occurrences are prevalent in the ``fp16`` mode, where an adaptive scaler is employed to ensure the preservation of small gradients. Previous experiments demonstrated a decrease in the frequency of ``skip`` events (wherein the scaler skips an update step and reduces the scale if Inf or NaN exists in gradients) upon the application of PGC, providing evidence that the pixel-wise design is more effective in stabilizing the training process.

---

### Official Review · Reviewer_EEyr · 2023-11-01

**Soundness:** 3 good
**Presentation:** 3 good
**Contribution:** 3 good
**Rating:** 6
**Confidence:** 2

**Summary:**

In the field of high-resolution 3D object generation, the limited availability of training data necessitates the widespread use of knowledge transfer techniques. However, in such methods (SDS), it is challenging to control gradients at the pixel level when applying pretrained models. To address this issue, this paper proposes a method called Pixel-wise Gradient Clipping (PGC) to effectively control gradients and incorporate them into existing 3D generation models. The proposed PGC offers a straightforward approach that can greatly aid in rendering high-quality, high-resolution objects. This paper proposes a reasonable approach within the context of previous research efforts in the actively studied field of 3D object generation to achieve high-quality results in high resolution. It introduces not only regularization for Score Distillation sampling but also Pixel-wise normalized gradient descent (PNGD) to preserve the details of textures. These techniques aim to generate high-resolution outputs with excellent quality.

**Strengths:**

This paper provides a clear and intuitive explanation of the proposed PGC and PNGD methods, showcasing high-quality results that align with these techniques. Additionally, considering the inherent ambiguity in defining metrics in the field, the paper presents indirect yet convincing numerical values through user studies. Furthermore, qualitative results are presented through experiments on mesh optimization, demonstrating significant performance improvements.

**Weaknesses:**

I think it would be better if there is an in-depth analysis regarding the gradient control aspect of the proposed method. It would be beneficial to have more thorough consideration and analysis on how the gradient is influenced by the proposed method compared to the vanilla model. Presenting a more comprehensive investigation and analysis in this regard would enhance the paper. It is disappointing that the stability of the gradient through PGC and PNGD is only demonstrated through the results without further analysis.

**Questions:**

1. I am curious if there were any side effects that arose from applying the proposed method to 3D object generation.
2. I am curious if there are any experimental results for more unusual exceptional samples that were tested for comparison evaluation.

---

> ### Author Response · Authors · 2023-11-21
>
> **Q1: In-depth analysis.**
>
> We acknowledge that we find enhancement in the experiments. However, we have tried our best to explain the phenomenon in Section 4 of the main paper.
> First from the failure of the vanilla method and partial success of latent gradients/linear approximated gradients (Columns (i)(ii)(iii) of Figure 2), we find the gradient issue. Then through the numerical value (Table below) and visualization of the gradients (Row (b) of Figure 2), we establish the noise assumption (Section 4.4.1). Based on the assumption and equations 8&10, we can effectively remove the noise through equation 11. Figure 2 also shows how these gradients affect the generation quality.
>
> To gain further insights, we analyze the distribution of per-pixel gradient norms through a histogram on all the images at different steps during one optimization on 2D experiments using SDXL (as the Row (a) of Figure 2). The table below illustrates our findings, revealing that 1.3% of pixel gradient norms surpass 1,000, predominantly falling outside the typical pixel value range of [0, 1]. Additionally, our investigation indicates that 99.8% of images contain at least one pixel with a gradient norm exceeding 10,000.
> | >1  | >10  | >100 | >1,000 |
> |  ----  | ----  | ----  | ----  |
> | 64\% |  17\% |  6\%  |  1.3\% |
>
> **Q2: Any side effects?**
>
> In the mesh optimization where an initial mesh is provided, we observe a consistent improvement using our method. We did find side effects in NeRF optimization from scratch: sometimes the shape may vanish at the beginning when we use SDS or VSD loss with PGC. However, Stable-DreamFusion and Fantasia3D use (normal+mask) as latent for SDS in the early training. This provides a more stable shape initialization. For other pipelines (e.g. ProlificDreamer), we can also skip the early training stage and directly apply our proposed PGC in the later training.
>
> **Q3: More results?**
>
> Yes. Thanks. We have now provided more results in the **separate supplementary file** (Figures 7, 8, 9).

---

> > ### Comment · Reviewer_EEyr · 2023-11-23
> >
> > I appreciated the author's rebuttal for my opinion. I considered all rebuttals from other reviewers. And I would like to keep my original rating.

---

### Author Response · Authors · 2023-11-21
**To all reviewers**

We thank all reviewers for their detailed review and constructive suggestions. We have revised the manuscript with **highlighted color**. We also replied to all the reviewers' questions one by one in detail and included experiment results in the **separate supplementary file**.

---

### Meta-Review · Area_Chair_YUMf · 2023-12-08

**Metareview:**

The paper proposed Pixel-wise Gradient Clipping (PGC) to effectively incorporate popular and powerful pretrained 2D image generative models such as latent diffusion models to generate 3D objects. The paper is addressing an important and fundamental problem, and can be used as an enabler for many 2D image models to be applied to 3D.

The reviewers are all positive about the paper, highlighting the strengths of the paper: it provides a clear and intuitive explanation of the proposed PGC and PNGD method, and showcases high-quality results aligning with the techniques (particularly through user studies and mesh optimization experiments); the paper is well-written, easy to follow, and adequately addresses the important goal of improving gradient flows in 3D models using 2D diffusion models; the proposed PGC is effective compared to various baselines, demonstrating potential for widespread application and future work in the field.

The reviewers also shared weaknesses: there is a notable lack of in-depth analysis on the gradient control aspect of the proposed method; insufficient consideration of how the gradient is influenced compared to the standard model; lacks a comprehensive ablation study, especially on pixel-wise NGD and GC, and does not adequately compare these with parameter-wise NGD and GC; insufficient experiments to substantiate the claim that PGC benefits existing SDS and LDM-based 3D generative models; some figures and descriptions, particularly Figure 2, are unclear, resulting in difficulties in understanding the experimental setup and findings.

The authors did a good job addressing the reviewers’ concerns in the rebuttal with proper new results/numbers or explanations. The reviewers were happy about the rebuttal and gave a unanimous score of marginal accept.

**Justification For Why Not Higher Score:**

While the experimental results are convincing, in-depth analysis is missing in many aspects. It would be great if the paper digs deeper into the theoretical analysis of gradient explosion for 3D objects and provides insights of the solution.

**Justification For Why Not Lower Score:**

All three reviewers gave (marginal) accept rating and the paper has potential to be influential in the 3D generative model space.

---

### Decision · Program_Chairs · 2024-01-16

Accept (poster)